# Factorizing Declarative and Procedural Knowledge in Structured, Dynamical Environments

**Anirudh Goyal[1], Alex Lamb [1], Phanideep Gampa [1,2], Philippe Beaudoin [3], Sergey Levine [4], Charles Blundell [5], Yoshua Bengio [1], Michael Mozer [6]**

## Abstract

Modeling a structured, dynamic environment like a video game requires keeping track of the objects and their states (*declarative* knowledge) as well as predicting how objects behave (*procedural* knowledge). Black-box models with a monolithic hidden state often fail to apply procedural knowledge consistently and uniformly, i.e., they lack *systematicity*. For example, in a video game, correct prediction of one enemy's trajectory does not ensure correct prediction of another's. We address this issue via an architecture that factorizes declarative and procedural knowledge and that imposes modularity within each form of knowledge. The architecture consists of active modules called *object files* that maintain the state of a single object and invoke passive external knowledge sources called *schemata* that prescribe state updates. To use a video game as an illustration, two enemies of the same type will share schemata but will have separate object files to encode their distinct state (e.g., health, position). We propose to use attention to determine which object files to update, the selection of schemata, and the propagation of information between object files. The resulting architecture is a drop-in replacement conforming to the same input-output interface as normal recurrent networks (e.g., LSTM, GRU) yet achieves substantially better generalization on environments that have multiple object tokens of the same type, including a challenging intuitive physics benchmark.

## 1    Introduction

An intelligent agent that interacts with its world must not only perceive objects but must also remember its past experience with these objects. The wicker chair in one's living room is not just a chair, it is the chair which has an unsteady leg and easily tips. Your keys may not be visible, but you recall placing them on the ledge by the door. The annoying fly buzzing in your left ear is the same fly you saw earlier which landed on the table.

Visual cognition requires a short-term memory that keeps track of an object's location, properties, and history. In the cognitive science literature, this particular form of state memory is often referred to as an *object file* (Kahneman et al., 1992), which we'll abbreviate as OF. An OF serves as a temporally persistent reference to an external object, permitting object constancy and permanence as the object and the viewer move in the world.

Complementary to information in the OF is abstract knowledge about the dynamics and behavior of an object. We refer to this latter type of knowledge as a *schema* (plural *schemata*), another term borrowed from the cognitive-science literature. The combination of OFs and schemata is sufficient to predict future states of object-structured environments, critical for planning and goal-seeking behavior.

To model a complex, structured visual environment, multiple OFs must be maintained in parallel. Consider scenes like a PacMan video-game screen in which the ghosts chase the PacMan, a public

---

[01] Mila, University of Montreal, [2] IIT BHU, Varanasi, [3] Waverly, [4] UC Berkeley, [5] Deepmind, [6] Google Research, Brain Team, Corresponding author: `anirudhgoyal9119@gmail.com`

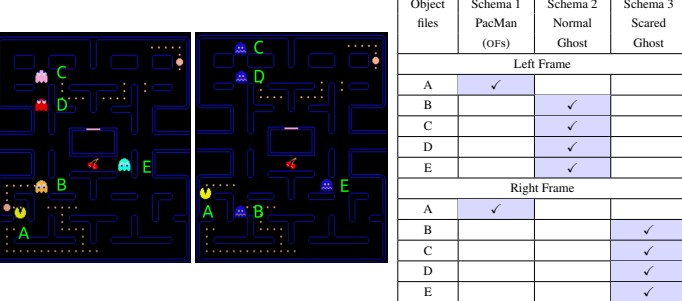

| Object files | Schema 1 PacMan (OFs) | Schema 2 Normal Ghost | Schema 3 Scared Ghost |
|---|---|---|---|
| Left Frame | | | |
| A | ✓ | | |
| B | | ✓ | |
| C | | ✓ | |
| D | | ✓ | |
| E | | ✓ | |
| Right Frame | | | |
| A | ✓ | | |
| B | | | ✓ |
| C | | | ✓ |
| D | | | ✓ |
| E | | | ✓ |

Figure 1: Two successive frames of PacMan, illustrating the factorization of knowledge. Each ghost is represented by a persistent OF (maintaining its location and velocity), but all ghosts operate according to one of two schemata, depending on whether the ghost is in a *normal* or *scared* state.

square or sports field in which people interact with one another, or a pool table with rolling and colliding balls. In each of these environments, multiple instances of the same object class are present; all operate according to fundamentally similar dynamics. To ensure systematic modeling of the environment, the same dynamics must be applied to multiple object instances. Toward this goal, we propose a method of separately representing the state of an individual object—via an OF—and how its state evolves over time—via a schema.

Object-oriented programming (OOP) provides a metaphor for thinking about the relationship between OFs and schemata. In OOP, each *object* is an instantiation of an object class and it has a self-contained collection of variables whose values are specific to that object and *methods* that operate on all instances of the same class. The relation between objects and methods mirrors the relationship between our OFs and schemata. In both OOP and our view of visual cognition, a key principle is the *encapsulation* of knowledge: internal details of objects (OFs) are hidden from other objects (OFs), and methods (schemata) are accessible to all and only objects (OFs) to which they are applicable.

The modularity of knowledge in OOP supports human programmers in writing code that is readily debugged, extended, and reused. We conjecture that the corresponding modularity of OFs and schemata will lead to neural-network models with more efficient learning and more robust generalization, thanks to appropriate disentangling and separation of concerns.

Modularity is the guiding principle of the model we propose, which we call SCOFF, an acronym for *schema / object-file factorization*. Like other neural net models with external memory (e.g., Mozer and Das, 1993; Graves et al., 2016; Sukhbaatar et al., 2015), SCOFF includes a set of slots which are each designed to contain an OF (Figure 2). In contrast to most previous external memory models, the slots are not passive contents waiting to be read or written by an active process, but are dynamic, modular elements that seek information in the environment that is relevant to the object they represent, and when critical information is observed, they update their states, possibly via information provided by other OFs. Event-based OOP is a good metaphor for this active process, where external events can trigger the action of objects.

As Figure 2 suggests, there is a factorization of *declarative* knowledge—the location, properties, and history of an object, as contained in the OFs—and *procedural* knowledge—the rules of object behavior, as contained in the schemata. Whereas declarative knowledge can change rapidly, procedural knowledge is more stable over time. This factorization allows any schema to be applied to any OF as

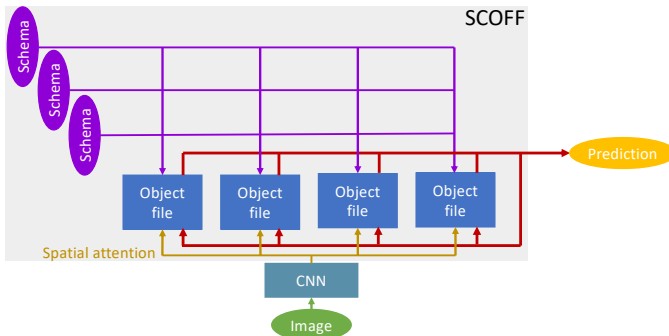

Figure 2: *Proposed SCOFF model.* Schemata are sets of parameters that specify the dynamics of objects. Object files (OFs) are active modules that maintain the time-varying state of an object, seek information from the input, and select schemata for updating, and transmit information to other object files. Through spatial attention, OFs compete and select different regions of the input.

deemed appropriate. The model design ensures *systematicity* in the operation of a schema, regardless of the slot to which an OF is assigned. Similarly, an OF can access any applicable schema regardless of which slot it sits in. Furthermore, a schema can be applied to multiple OFs at once, and multiple schemata could be applied to an OF (e.g., Figure 1). In OOP, systematicity is similarly achieved by virtue of the fact that the same method can be applied to any object instantiation and that multiple methods exist which can be applied to an object of the appropriate type.

Our key contribution is to demonstrate the feasibility and benefit of factorizing declarative knowledge (the location, properties, and history of an object) and procedural knowledge (the way objects behave). This factorization enforces not only an important form of systematicity, but also of exchangeability: the model behaves exactly the same regardless of the assignment of schemata to schemata-slots and the assignment of objects to OF-slots, i.e., the neural network operates on a *set* of objects and a *set* of schemata. With this factorization, we find improved accuracy of next-state prediction models and improved interpretability of learned parameters.

## 2 THE SCHEMATA / OBJECT-FILE FACTORIZATION (SCOFF) MODEL

SCOFF, shown in Figure 2, is an architectural backbone that supports the separation of procedural and declarative knowledge about dynamical entities (objects) in an input sequence. The input sequence $\{x_1, \ldots, x_t, \ldots, x_T\}$, indexed by time step $t$ is processed by a neural encoder (e.g., a fully convolutional net for images) to obtain a deep embedding, $\{z_1, \ldots, z_t, \ldots, z_T\}$, which then drives a network with $n_f$ OFs and $n_s$ schemata.

OFs are active processing components that maintain and update their internal state. Essentially, an OF is a layer of GRU (Chung et al., 2014) or LSTM (Hochreiter and Schmidhuber, 1997) units with three additional bits of machinery, which we now describe.

1. Our earlier metaphor identifying OFs in SCOFF with objects in OOP is apropos in the sense that OFs are *event driven*. OFs operate in a temporal loop, continuously awaiting relevant input signals. Relevance is determined by an attention-based soft competition among OFs: the current input serves a key that is matched to a query generated from the state of each OF; based on the goodness of match, each OF is provided with a value extracted from the input.

2. Each OF performs a one-step update of its state layer of GRU or LSTM units, conditioned on the input signal received. The weight parameters needed to perform this update, which we will denote generically as $\theta$, are not—as in a standard GRU or LSTM—internal to the layer but rather are provided externally. Each schema $j$ is nothing more than a set of parameters $\theta_j$ which can be plugged into this layer. SCOFF uses a key-value attention mechanism to perform Gumbel-based hard selection of the appropriate schema (parameters).

3. OFs may pass information to other OFs (analogous to arguments being passed to a method in OOP), again using a soft attention mechanism by which each OF queries all other OFs. Keys provided by the other OFs are matched to the query, and a soft selection of the best matching OFs determines the weighting of values transmitted by the other OFs.

This operation cycle ensures that OFs can update their state in response to both the external input and the internal state comprised of all the OFs' contents. This updating is an extra wrapper around the ordinary update that takes place in a GRU or LSTM layer. It provides additional flexibility in that (1) The external input is routed to OFs contingent on their internal state, (2) OFs can switch their dynamics from one time step to the next conditioned on their internal state, (3) OFs are modular in that they do not communicate with one another except via state-dependent selective message passing.

OFs are placeholders in that a particular OF has no weight parameter specific to that OF. Instead, the parameters are provided from two sources: either the schemata or a pool of generic parameters shared by the $n_f$ OFs. This generic parameter pool is used to implement key-value attention over the input, the schemata, and communication among OFs. The sharing of schemata ensures systematicity; the sharing of the generic parameter pool ensures *exchangeability*—model behavior is unaffected by the assignment of object instances to OF slots.

### 2.1 SCOFF SPECIFICS

---

**Algorithm 1** SCOFF model

---

**Input:** Current sequence element, $\boldsymbol{x}_t$ and previous OF state, $\{\boldsymbol{h}_{t-1,k}|\,k \in \{1,\ldots,n_f\}\}$

***Step 1: Process image by position $p$ with fully convolutional net***
- $\boldsymbol{c}_p = [\mathrm{CNN}(\boldsymbol{x}_t)]_p$
- $\boldsymbol{a}_p = [\boldsymbol{c}_p\ \boldsymbol{e}_p]$   *(concatenate encoding of position to CNN output)*

***Step 2: Soft competition among OFs to select regions of the input to process***
- $\boldsymbol{q}_k = \boldsymbol{W}^q \boldsymbol{h}_{t-1,k}$
- $s_{k,p} = \mathrm{softmax}_k\!\left(\frac{\boldsymbol{q}_k^{\mathrm{T}}\boldsymbol{\kappa}_p}{\sqrt{d_e}}\right)$, where $\boldsymbol{\kappa}_p = (\boldsymbol{a}_p \boldsymbol{W}^e)^{\mathrm{T}}$
- $\boldsymbol{z}_k = \sum_p s_{k,p}\boldsymbol{v}_p$   where $\boldsymbol{v}_p = \boldsymbol{a}_p \boldsymbol{W}^v$   $\forall\,k \in \{1,\ldots,n_f\}$

***Step 3: OFs pick the most relevant schema and update***
- $\widetilde{\boldsymbol{h}}_{t,k,j} = \mathrm{GRU}_{\boldsymbol{\theta}_j}(\boldsymbol{z}_k, \boldsymbol{h}_{t-1,k})$   $\forall\,k \in \{1,\ldots,n_f\}, j \in \{1,\ldots,n_s\}$
- $\widetilde{\boldsymbol{q}}_k = \boldsymbol{h}_{t-1,k}\widetilde{\boldsymbol{W}}^q$
- $i_k = \mathrm{argmax}_j\left(\widetilde{\boldsymbol{q}}_k^{\mathrm{T}}\widetilde{\boldsymbol{\kappa}}_{k,j} + \gamma\right)$, where $\widetilde{\boldsymbol{\kappa}}_{k,j} = (\widetilde{\boldsymbol{h}}_{t,k,j}\widetilde{\boldsymbol{W}}^e)^{\mathrm{T}}$ and $\gamma \sim \mathrm{Gumbel}(0,1)$
- $\boldsymbol{h}_{t,k} = \widetilde{\boldsymbol{h}}_{t,k,i_k}$

***Step 4: Soft competition among OFs to transmit relevant information to each OF***
- $\widehat{\boldsymbol{q}}_k = \boldsymbol{h}_{t-1,k}\widehat{\boldsymbol{W}}^q$   $\forall k\{1,\ldots,n_f\}$
- $s_{k,k'} = \mathrm{softmax}_{k'}\left(\frac{\widehat{\boldsymbol{q}}_k^{\mathrm{T}}\widehat{\boldsymbol{\kappa}}_{k'}}{\sqrt{d_e}}\right)$ where $\widehat{\boldsymbol{\kappa}}_{k'} = (\boldsymbol{h}_{t,k'}\widehat{\boldsymbol{W}}^e)^{\mathrm{T}}$   $\forall\,k,\ k' \in \{1,\ldots,n_f\}$
- $\boldsymbol{h}_{t,k} \leftarrow \boldsymbol{h}_{t,k} + \sum_{k'} s_{k,k'}\widehat{\boldsymbol{v}}_{k'}$ where $\widehat{\boldsymbol{v}}_{k'} = \boldsymbol{h}_{t,k'}\widehat{\boldsymbol{W}}^v$   $\forall k \in \{1,\ldots,n_f\}$

---

Algorithm 1 provides a precise specification of SCOFF broken into four steps. Step 1 is external to SCOFF and involves processing an image input to obtain a deep embedding. The processing is performed by a fully Convolutional Neural Net (CNN) that preserves positional information (typically $64 \times 64$ in our simulations), and for each position $p$ encodes the processed input $\boldsymbol{c}_p$ and concatenates a learned encoding of position, $\boldsymbol{e}_p$ to produce a position-specific hidden state. (SCOFF can also handle non-visual inputs, such as vectors or discrete tokens; in this case the CNN is replaced by an MLP.) The subsequent core steps are as follows.

***Step 2: Soft competition among OFs to select regions of the input to process.*** The state of each OF $k$, $\boldsymbol{h}_{t-1,k}$, is used to form a query, $\boldsymbol{q}_k$, which determines the input positions that it will attend to. The query is matched to a set of position-specific input keys, $\boldsymbol{\kappa}_{k,p}$ for position $p$, producing a position-specific match score, $s_{k,p}$. Soft position-specific competition among the OFs results in the OFs selecting complementary image regions to attend to. The contents of the attended positions are combined yielding an OF-specific input encoding, $\boldsymbol{z}_k$.

***Step 3: OFs pick the most relevant schema and update.*** OF $k$ picks one schema via attention as follows. OF $k$ binds to *each* schema $j$, and then performs a hypothetical update, yielding $\widetilde{\boldsymbol{h}}_{t,k,j}$. In experiments below, the OF state is maintained by a GRU layer, and schema $j$ is a parameterization of the GRU, denoted $\mathrm{GRU}_{\boldsymbol{\theta}_j}$, which determines the update. The previous state of the OF, $\boldsymbol{h}_{t-1,k}$ serves as a query in key-value attention against a key derived from the hypothetical updated state, $\widetilde{\boldsymbol{h}}_{t,k,j}$. The schema $i_k$ corresponding to the best query-key match for OF $k$ is used to update OF $k$'s state. Selection is based on the straight-through Gumbel-softmax method (Jang et al., 2016), which makes a hard choice during forward propagation and during backward propagation, it considers a softened version of the output to permit gradient propagation to non-selected schemata.

***Step 4: Soft competition among OFs to transmit relevant information to each OF.*** This step allows for interactions among OFs. Each OF $k$ queries other OFs for information relevant to its update as follows. The OF's previous state, $\boldsymbol{h}_{t-1,k}$, is used to form a query, $\widehat{\boldsymbol{q}}_k$, in an attention mechanism against a key derived from the new state of each other OF $k'$, $\widehat{\boldsymbol{\kappa}}_{k'}$, and softmax selection ($s_{k,k'}$) is used to obtain weighted information from other OFs, $\widehat{\boldsymbol{v}}'_k$, into OF $k$'s state.

*Number of Parameters.* SCOFF can be used as a drop-in replacement for a LSTM/GRU layer. There is a subtlety that must be considered for successful integration. If the total size of the hidden state is kept the same, integrating SCOFF dramatically reduces the total number of recurrent parameters in

the model because of its block-sparse structure. The majority of SCOFF parameters are the schemata, $\{\boldsymbol{\theta}_j | j \in \{1, \ldots, n_s\}\}$. The remaining parameters are those of query ($\boldsymbol{W}^q$, $\widetilde{\boldsymbol{W}}^q$, $\widehat{\boldsymbol{W}}^q$), key ($\boldsymbol{W}^e$, $\widetilde{\boldsymbol{W}}^e$, $\widehat{\boldsymbol{W}}^e$), and value $\boldsymbol{W}^v$, $\widehat{\boldsymbol{W}}^v$) functions. Note that these linear functions could be replaced by nonlinear functions. Its also interesting to note that SCOFF actually has far fewer parameters than other modular architectures (e.g., RIMs, Recurrent Entity Networks, Neural Module Networks) when $n_s < n_f$ (which holds true for most of our experiments), because of the potential one-to-many mapping between schemata and OFs. Optionally, at *Step 2*, during training instead of activating all the OFs, we can use a sparse attention, to only activate a subset of OFs that are relevant at that time step $t$, and only update the state of the activated OFs. We note that this is not specific to the proposed method, but the performance of the method can be improved by only selectively activating the relevant OFs. We also note that the different schemata in the proposed method are only used to update the state of OFsand hence is agnostic as to how one obtains a set of OFs. Any other method for entity extraction (Burgess et al., 2019; Greff et al., 2019) can be used for obtaining a set of OFs. Similarly, other methods for decoding which preserves equivariance (Watters et al., 2019) among different OFscan be used for decoding the contents of different OFs.

## 3 RELATED WORK

**CNNs**. SCOFF applies the same knowledge (schemata) to multiple OFs, yielding systematicity. Similarly, a CNN is a highly restrictive instantiation of this same notion, where knowledge—in the form of a convolutional filter—is applied uniformly to every location in an image, yielding equivariance. SCOFF is a more flexible architecture than a CNN in that OFs are defined by abstract notion of objects not a physical patch of an image, and schemata are applied dynamically and flexibly over time, not in a fixed, rigid manner as the filters in a CNN.

**Memory Networks**. A variety of existing models leverage an external slot-based, content-addressible memory (e.g., Graves et al., 2016; Sukhbaatar et al., 2015). The memory slots are passive elements that are processed by a differentiable neural controller such as an LSTM. Whereas traditional memory networks have many dumb memory cells and one smart controller, SCOFF has many smart memory cells—the OFs—which also act as local controllers. However, SCOFF shares the notion with memory networks that the same knowledge is applied systematically to every cell.

**Relational RNN (RMC)**. The RMC (Santoro et al., 2018) has a multi-head attention mechanism which allows it to share information between multiple memory locations. RMC is like Memory Networks in that dynamics are driven by a central controller: memory is used to condition the dynamics of an RNN.

**Recurrent Entity Networks**. Henaff et al. (2016) describe a collection of recurrent modules that update independently and in parallel in response to each input in a sequence. The module outputs are integrated to form a response. It shares a modular architecture with SCOFF, but the modules have a fixed function and do not directly communicate with one another. This earlier work focused on language not images.

**Recurrent Independent Mechanisms (RIMs)**. RIMs (Goyal et al., 2019) are a key inspiration for our work. RIMs are a modular neural architecture consisting of an ensemble of dynamical components which have sparse interactions through the bottleneck of attention. Each RIM module has at its core an LSTM layer (Hochreiter and Schmidhuber, 1997). A RIM module is much like our OF. Both are meant to be dynamical entities with a time-evolving state. However, in contrast to OFs in SCOFF, RIM modules operate according to fixed dynamics and each RIM module is specialized for a particular computation. RIM modules are thus not interchangeable.

**Neural Module Networks**. A modular module network (Jacobs et al., 1991; Bottou and Gallinari, 1991; Ronco et al., 1997; Reed and De Freitas, 2015; Andreas et al., 2016; Rosenbaum et al., 2017; Fernando et al., 2017; Shazeer et al., 2017; Kirsch et al., 2018; Rosenbaum et al., 2019) has an architecture which is composed dynamically from several neural modules, where each module is meant to perform a distinct function. Each module has a fixed function, unlike our OFs, and modules are applied one at a time, in contrast to our OFs, which can update and be used for prediction in parallel.

## 4  METHODOLOGY

SCOFF is a drop-in replacement for a standard LSTM or GRU layer, conforming to the same input-output interface. Because of their interchangeability, we compare SCOFF to LSTM and GRUs. We also compare SCOFF to two alternative modular architectures: *RMC*, a memory based relational recurrent model with attention between memory elements and hidden states (Santoro et al., 2018), and *Recurrent Independent Mechanisms (RIMs)*, a modular memory based on a single layered recurrent model with attention modulated input and communication between modules (Goyal et al., 2019).

In all simulations we present, the input is a video sequence, each frame of which is preprocessed by a CNN backbone. We consider two task types: video prediction and reinforcement learning (RL). For video prediction, the simulation output is a prediction of the next frame in the sequence. For RL, the output is an action choice. In both cases, SCOFF's internal state is mapped to an output in the following manner. The state of each OF is remapped by a siamese network that transforms the state. (By siamese network, we mean that every OF is remapped by the same function.) The transformed state is concatenated and used by an attention-based mechanism operating over the OF slots to ensure exchangeability of the OFs. In the case of video prediction, a deconvolutional backbone yields an image as output; in the case of reinforcement learning, the output is a distribution over actions.

The heart of the model is a single SCOFF layer, consisting of $n_f$ OFs and $n_s$ schemata. Most simulation details are contained in section A of the Appendix, but we summarize some key points here. Unless otherwise indicated, we always train in an end-to-end fashing using the Adam (Kingma and Ba, 2014) optimizer with a learning rate of 0.0001 and momentum of 0.9. As a default, we use $n_f = 6$ and $n_s = 4$, except where we are specifically exploring the effects of manipulating these hyperparameters. We include more experimental results in the Appendix, and we will release the code.

## 5  EXPERIMENTS

Our experiments address the following questions. (1) Does SCOFF successfully factorize knowledge into OFs and schemata? (2) Do the learned schemata have semantically meaningful interpretations? (3) Is the factorization of knowledge into object files and schemata helpful in downstream tasks? (4) Does SCOFF outperform state-of-the-art approaches, both modular and non-modular, which lack SCOFF's strong inductive bias toward systematicity and knowledge factorization?

We begin with a series of experiments involving greyscale synthetic video sequences consisting of a single ball-like object moving over time according to switchable dynamics. We model this scenario with SCOFF having a single ($n_f = 1$) OF and the number of schemata matching the number of different types of dynamics.

***Single object with fixed dynamics.*** To demonstrate that the proposed model is able to factorize knowledge, we consider video scenes in which a single object, starting in a random location, has dynamics that cause it to either (a) accelerate in a particular direction, (b) move at a constant velocity in a particular direction, or (c) take a random walk with a constant velocity. Following training, SCOFF can predict trajectories after being shown the first few frames. It does so by activating a schema that has a one-to-one correspondence with the three types of dynamics (Figure 3, left panel), leading

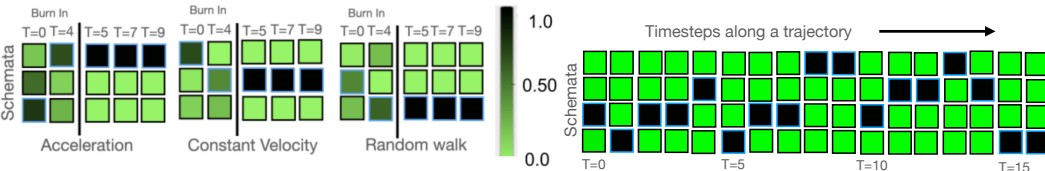

Figure 3: [*left panel*] Single object sequences with three possible dynamics of motion. After about five video frames (burn-in period), SCOFF locks into the type of motion and activates a corresponding schema to predict future states. Relative activation of the three schemata indicated by the color bar. The selected schema indicated by the faint blue border. [*right panel*] *RL Maze task*. The agent learns to navigate a maze of randomly interconnected rooms to find a key. Array indicates schema activation over a sequence of steps along a particular trajectory.

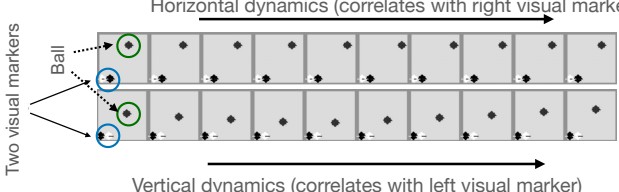

Figure 4: *Switching dynamics task.* Sequence of steps showing horizontal (top row) and vertical (bottom row) dynamics of a ball. Frames contain visual markers that indicate the current dynamics.

to *interpretable semantics* and a clean factorization of knowledge. For additional details, refer to Appendix C.

***Single object with switching dynamics.*** In this experiment, video scenes contain a single object, starting in a random location, operating according to one of two dynamics: vertical and horizontal motion. In contrast to the previous experiment, the object can switch dynamics in the course of a sequence. The current dynamics are indicated by markers in the image (see Figure 4) which SCOFF must learn to detect in order to select the corresponding schema. In this experiment, we find that SCOFF is able to learn a one-to-one correspondence with two types of dynamics, yielding high-precision prediction of the next frame, even on the first frame after a switch.

***Single object with switching dynamics in an RL paradigm.*** In this experiment, we use the partially-observable GotoObjMaze environment from Chevalier-Boisvert et al. (2018), in which an agent must learn to navigate to navigate a 2D multi-room grid world to locate distinct objects such as a key. The world consists of $6 \times 6$ square rooms that are randomly connected through doors to form a $3 \times 3$ maze of rooms. The agent's view of the world is an egocentric $5 \times 5$ patch. Our experiment involves $n_s = 4$ schemata whose activation pattern over time steps is shown in the right panel of Figure 3. The schema activation pattern is interpretable; for example, schema 4 is triggered when the 'key' is in the agent's field of view (see Appendix, Figure 13 and Section D). To obtain quantitative evaluations, we test transfer by increasing the room size to $10 \times 10$ during testing. SCOFF is able to successfully reach its goal on 82% of trials, whereas a GRU baseline succeeds only 56% of trials. Even an overparameterized GRU, which matches SCOFF in number of free parameters, still succeeds less often—on 74% of trials. All models are trained for an equal number of time steps. To summarize, the factorization of knowledge in SCOFF leads not only to better next-step prediction but also improve performance on a downstream control task.

***Multiple objects with multiple dynamics.*** We now turn from single-object sequences to sequences involving multiple objects which operate according complex dynamics. We consider a bouncing-balls environment in which multiple balls move with billiard-ball dynamics (Van Steenkiste et al., 2018). The dataset consists of 50,000 training examples and 10,000 test examples showing ∼50 frames of either 4 solid same-color balls bouncing in a confined square geometry with different masses corresponding to their radii (*4Balls*), 6-8 same-color balls bouncing in a confined geometry (*678Balls*), 3 same-color balls bouncing in a confined geometry with a central occluder (*Curtain*), or balls of four different colors (*Colored 678Balls*). Although each ball has a distinct state (position, velocity, and possibly color), they share the same underlying dynamics of motion and collision. We expected that SCOFF would learn to dissect these dynamics, e.g., by treating collisions, straight-line motion, interactions with the walls, and moving behind an occluder as distinct schemata. We use the encoder and decoder architecture of (Van Steenkiste et al., 2018) for all models. SCOFF has $n_f = 4$ for 4Balls and Curtain, and $n_f = 8$ for 678Balls and Coloured678Balls, and $n_s = 4$ for all simulations. All models are trained for 100 epochs. As shown in Figure 5, SCOFF achieves dramatic improvements in successfully predicting 10- and 30-step iterated predictions relative to GRUs, and RIMs. (For further details, see Appendix Section C.) We omitted RMC (Santoro et al., 2018) from Figures 5a-d because RIMs performs strictly better than RMC in these environments (Goyal et al., 2019), but have included RMC in the Appendix.

***Increasing the number of schemata.*** Our previous experiment used $n_s = 4$ for all simulations. In order to study what happens if SCOFF has access to a large number of schemata, we perform an experiment with $n_s = 10$. For the curtain task, we found that only three schemata were being used, and performance was about the same as when training with $n_s = 3$ or $n_s = 4$. For 678Balls, we found that by the end of training only four schemas were being used, and performance is about the same as when training with $n_s = 4$. Thus, providing SCOFF with excess resources leads to some not being used which is a waste of computation and memory but does not lead to suboptimal performance.

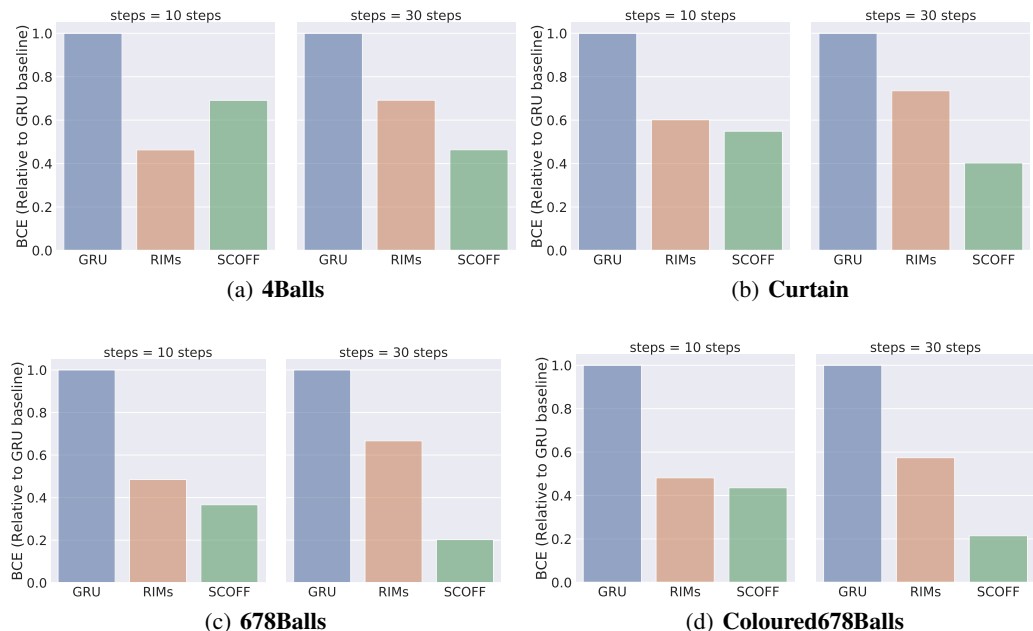

Figure 5: *Bouncing ball motion.* Error relative to GRU baseline on 10- and 30-frame video prediction of multiple-object video sequences Predictions are based on 15 frames of ground truth. The advantage of SCOFF is amplified as the number of balls increases (4Balls versus 678Balls) and as predictions require greater look ahead (10 versus 30 frames).

|  | Block O1 (Disappear) | Block O2 (Shape Change) | Block O3 (Teleport) |
|---|---|---|---|
| RMC (Santoro et al., 2018) | $0.43 \pm 0.05$ | $0.39 \pm 0.03$ | $0.46 \pm 0.02$ |
| Intphys (Riochet et al., 2019) | 0.52 | 0.52 | 0.51 |
| SCOFF (ours) | $0.34 \pm 0.05$ | $0.35 \pm 0.05$ | $0.42 \pm 0.02$ |

Table 1: *Results on the IntPhys benchmark.* Relative classification error of unrealistic physical phenomena (Riochet et al., 2019) for three models, demonstrating benefits of SCOFF in scenes with significant occlusions ("Occluded"). This particular benchmark has three subsets, and for our experiments we evaluate the proposed model on the "occlusion" subset of the task. The three columns correspond to 3 different types of occlusions. Lower is better. Average taken over 3 random seeds.

For example, one might have been concerned about overfitting with too many resources. However, with $n_s = 3$ schemata, we never observe unused schemata, suggesting that the model does not have difficulty using the resources we provide to it. That is, there are no 'dead' schemata that fail to be trained due to local optima.

***Modeling physical laws in a multi-object environment.*** Modeling a physical system, such as objects in a world obeying laws of gravity and momentum, requires factorization of state (object position, velocity) and dynamics. All objects must obey the same laws while each object must maintain its distinct state. We used the Intuitive Physics Benchmark (Riochet et al., 2019) in which balls roll behind a brick wall such that they are briefly occluded. The training data is constructed such that the examples are all physically realistic. The test set consists of both sequences that behave according to the laws used to synthesize the training data and sequences that follow unrealistic physical laws. We test three forms of unrealistic physics: balls disappearing behind the wall (O1 task), balls having their shape change for no reason (O2 task), and balls teleporting (O3 task). The Benchmark has three subsets of experiments, and we chose the challenging subset with significant occlusions. We trained models to perform next state prediction on the training set and we use model likelihood (Riochet et al., 2019) to discriminate between realistic and unnatural test sequences. Further details are in Appendix E. As Table 1 indicates, SCOFF significantly outperforms two competitors on all three tasks.

## 6 CONCLUSIONS

Understanding the visual world requires interpreting images in terms of distinct independent physical entities. These entities have persistent intrinsic properties, such as a color or velocity, and they have dynamics that transform the properties. We explored a mechanism that is able to factorize declarative knowledge (the properties) and procedural knowledge (the dynamics). Using attention, our SCOFF model learns this factorization into representations of entities—OFs—and representations of how they transform over time—schemata. By applying the same schemata to multiple OFs, SCOFF achieves systematicity of prediction, resulting in significantly improved generalization performance over state-of-the-art methods. It also addresses a fundamental issue in AI and cognitive science: the distinction between *types* and *tokens*. SCOFF is also interpretable, in that we can identify the binding between schemata and entity behavior. The factorization of declarative and procedural knowledge has broad applicability to a wide variety of challenging deep learning prediction tasks.

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

# Appendix

## Table of Contents

## A  IMPLEMENTATION DETAILS AND HYPERPARAMETERS

The model setup consists of three main components: an encoder, the process of interaction between object files and schemata followed by a decoder. The images are first processed by an encoder, which is parameterized as a CNN. For experiments containing multiple entities, we use a spatial attention (i.e., output of the CNN preserves the spatial information) such that different OFs can attend to different spatial regions.

**Resources Used.** It takes about 2 days to train the proposed model on bouncing ball task for 100 epochs on V100 (32G). We did not do any hyper-parameter search specific to a particular dataset (i.e 4Balls or 678Balls or Curtain Task). We ran the proposed model for different number of schemata (i.e 2/4/6). Similarly, it takes about 3 days to run for 20M steps for the Reinforcement learning task.

## B  ADDING TASK

We analyzed the proposed method on the adding task. This is a standard task for investigating recurrent models (Hochreiter and Schmidhuber, 1997). The input consists of two co-occuring sequences: 1) N numbers $(a_0 \cdots a_{N-1})$ sampled independently from $U[0, 1]$, 2) an index $i_0$ in the first half of the sequence, and an index $i_1$ in the second half of the sequence together encoder as a one hot sequences. The target output is $a_{i_0} + a_{i_1}$. As shown in figure 6 (a), we can clearly observe the factorisation of procedural knowledge into two schemata effectively, one schema is triggered when an operand is encountered and the other when non-operand is encountered.

**Generalization Result:**   For demonstrating the generalization capability of SCOFF, we consider a scenario where the models are trained to add a mixture of two and four numbers from sequences of length 50. They are evaluated on adding variable number (2-10) of numbers on 200 length sequences. As shown in Table 1, we note better generalization when using SCOFF. The dataset consists of 50,000 training sequences and 20,000 testing sequences for each different number of numbers to add.

All the models are trained for 100 epochs with a learning rate of 0.001 using the Adam optimizer. We use 300 as the hidden dimension for both the LSTM baseline and the LSTM's in RIMS, SCOFF. Table 2 lists the different hyperparameters used for training SCOFF.

Table 2: Hyperparameters for the adding generalization task

| Parameter | Value |
|---|---:|
| Number of object files ($n_f$) | 5 |
| Number of schemata ($n_s$) | 2 |
| Optimizer | Adam(Kingma and Ba, 2014) |
| learning rate | $1 \cdot 10^{-2}$ |
| batch size | 64 |
| Inp keys | 64 |
| Inp Values | 60 |
| Inp Heads | 4 |
| Inp Dropout | 0.1 |
| Comm keys | 32 |
| Comm Values | 32 |
| Comm heads | 4 |
| Comm Dropout | 0.1 |

| Number of Values | LSTM | RIMS | SCOFF |
|:---:|:---:|:---:|:---:|
| 2 | 0.8731 | 0.0007 | 0.0005 |
| 3 | 1.3017 | 0.0009 | 0.0007 |
| 4 | 1.6789 | 0.0014 | 0.0013 |
| 5 | 2.0334 | 0.0045 | 0.0030 |
| 8 | 4.8872 | 0.0555 | 0.0191 |
| 9 | 7.3730 | 0.1958 | 0.0379 |
| 10 | 11.3595 | 0.8904 | 0.0539 |

Table 3: **Adding Task:** Mean test set error on 200 length sequences with number of numbers to add varying among $\{2, 3, 4, 5, 8, 9, 10\}$. The models are trained to add a mixture of two and four numbers from sequences of length 50.

## C  BOUNCING BALL

The dataset consists of 50,000 training examples and 10,000 test examples showing ∼50 frames of either 4 solid balls bouncing in a confined square geometry (*4Balls*), 6-8 balls bouncing in a confined geometry (*678Balls*), 3 balls bouncing in a confined geometry with an occluded region (*Curtain*), or balls of different colors (*Colored 678Balls*). The balls have distinct states (and hence distinct object files) but share underlying procedures (schema), which we aim to capture using SCOFF.

We trained baselines as well as proposed model for about 100 epochs. . We also provide the rollouts predicted by the models in the Figures 8, 10, 9, 11, 12.

As part of future work, We also provide a scenario where instead of activating all the OF only a subset of the OF that are most relevant to the input based on attention scores are activated. We denote the original model as SCOFF-ab, and the ablation of only activating a subset of the OF as SCOFF.

Comparison to the RMC baseline is shown in fig. 7.

## D  BABYAI: REINFORCEMENT LEARNING

We use the GotoObjMaze environment (i.e MiniGrid-GotoObjMaze-v0) from (Chevalier-Boisvert et al., 2018). Here, the agent has to navigate to an object, the object may be in another room. We use exactly the same RL setup as in (Chevalier-Boisvert et al., 2018) except we extend the setup in BabyAI to only apply RGB images of the world rather than symbolic state representations, and hence making the task much more difficult. Hyper-parameters for this task are listed in Tab. 5.

In this environment, the agent is expected to navigate a 3x3 maze of 6x6 rooms, randomly inter-connected by doors to find an object like "key". Here we use only one object file, but different number

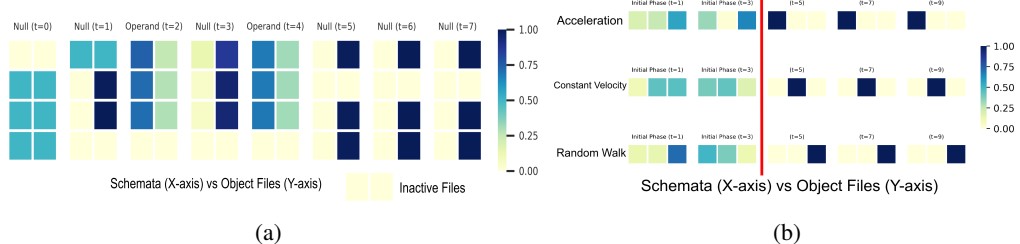

|     |     |
| :-: | :-: |
| (a) | (b) |

Figure 6: (a) OF ($n_f = 4$) vs Schemata ($n_s = 2$) activation for an example of length 8 of the adding task. "Null" refers to the elements other than the operands on which the addition is to be performed. The figure shows the affinity of each OF to use a particular schema. Each row corresponds to a particular OF, and column represents a particular schema (dark color shows high affinity of an OF toward a particular schema). As shown in the figure, the active OFs trigger Schema 1 when an operand is encountered, and Schema 2 when a "Null" element is encountered. (b) Here, we have a single OF, and that can follow three different dynamics. We found that our method is able to learn these 3 different modes once it's passed an initial phase of uncertainty.

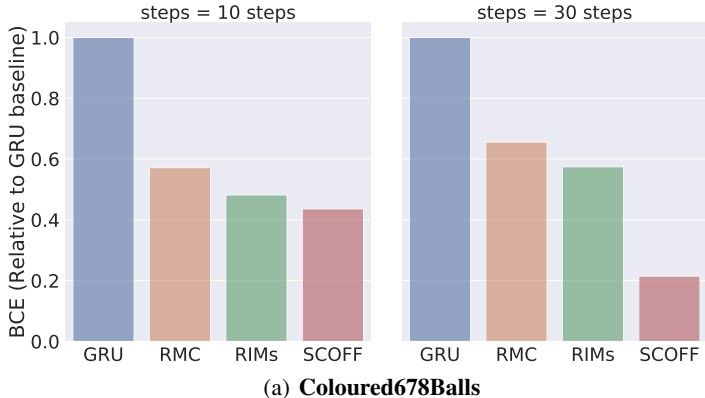

(a) **Coloured678Balls**

Figure 7: *Bouncing ball motion.* Error relative to GRU baseline on 10- and 30-frame video prediction of multiple-object video sequences Predictions are based on 15 frames of ground truth.

of schemas (4 in this example). If we look at the object files (vs) schemata, schemata 4 is being triggered when the "key" is in agent's view as shown in fig. 13.

## E  INTUITIVE PHYSICS

We use the similar training setup as (Riochet et al., 2019). Hyper-parameters related to the proposed method are listed in Tab. 6.

## F  SWITCHING DYNAMICS

We consider a scenario where a ball follows one of the two dynamics, horizontal and vertical oscillations at a given time. We limit the number of switches between the dynamics followed to be one. An example ground truth trajectory is given in figure 14. Here we run another experiment, where we have more number of OFs as compared to number of entities or objects, and then we investigate if we are still able to have factorization of different dynamics in different schemata.

Table 4: Hyperparameters for the bouncing balls task

| Parameter | Value |
|---|---:|
| Number of object files ($n_f$) | 4 |
| Number of schemata ($n_s$) | 2/4/6 |
| Size of Hidden state of object file | 100 |
| Optimizer | Adam(Kingma and Ba, 2014) |
| learning rate | $1 \cdot 10^{-4}$ |
| batch size | 64 |
| Inp keys | 64 |
| Inp Values | 100 |
| Inp Heads | 1 |
| Inp Dropout | 0.1 |
| Comm keys | 32 |
| Comm Values | 32 |
| Comm heads | 4 |
| Comm Dropout | 0.1 |

Table 5: Hyperparameters for BabyAI

| Parameter | Value |
|---|---:|
| Number of object files ($n_f$) | 1 |
| Number of schemata ($n_s$) | 2/4/6 |
| Size of Hidden state of object file | 510 |
| Optimizer | Adam(Kingma and Ba, 2014) |
| Learning rate | $3 \cdot 10^{-4}$ |
| Inp keys | 64 |
| Inp Values | 256 |
| Inp Heads | 4 |
| Inp Dropout | 0.1 |
| Comm keys | 16 |
| Comm Values | 32 |
| Comm heads | 4 |
| Comm Dropout | 0.1 |

## F.1 EXPERIMENT SETUP

The dataset consists of 10,000 trajectories of 51 length with the switching between dynamics happening at the middle of the trajectory.

Table 6: Hyperparameters for IntPhys benchmark

| Parameter | Value |
|---|---:|
| Number of object files ($n_f$) | 6 |
| Number of schemata ($n_s$) | 4/6 |
| Optimizer | Adam(Kingma and Ba, 2014) |
| learning rate | $3 \cdot 10^{-4}$ |
| batch size | 64 |
| Inp keys | 64 |
| Inp Values | 85 |
| Inp Heads | 4 |
| Inp Dropout | 0.1 |
| Comm keys | 32 |
| Comm Values | 32 |
| Comm heads | 4 |
| Comm Dropout | 0.1 |

We use the same architecture for encoder as well as decoder as in (Van Steenkiste et al., 2018). At each time step, we give the last five frames stacked across the channels as input to the encoder.

Table 7: Hyperparameters for the switching dynamics task

| Parameter | Value |
|---|---|
| Number of object files ($n_f$) | 6 |
| Number of schemata ($n_s$) | 2 |
| Optimizer | Adam(Kingma and Ba, 2014) |
| learning rate | $1 \cdot 10^{-4}$ |
| batch size | 50 |
| Inp keys | 64 |
| Inp Values | 64 |
| Inp Heads | 4 |
| Inp Dropout | 0.1 |
| Comm keys | 32 |
| Comm Values | 32 |
| Comm heads | 4 |
| Comm Dropout | 0.1 |

## F.2 THE FACTORISATION OF DYNAMICS INTO SCHEMATA

As shown in figure 15, SCOFF is effective in factorising the two different dynamics into two different schemata, even if the order of dynamics followed is different.

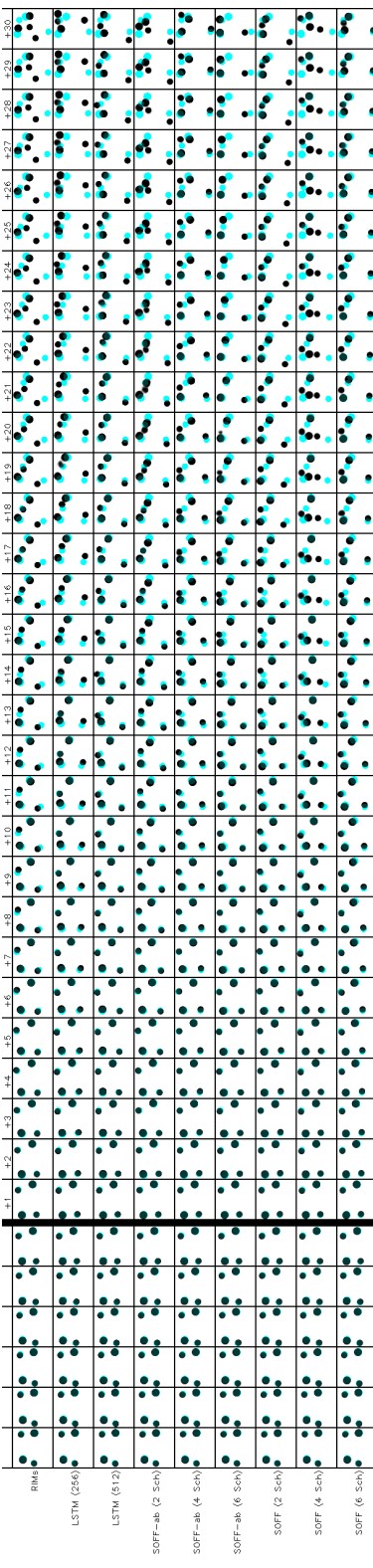

Figure 8: **Rollout for 4Balls.** In all cases, the first 10 frames of ground truth are fed in (last 6 shown) and then the system is rolled out for the next 30 time steps. In the predictions, the transparent blue shows the ground truth, overlaid to help guide the eye.

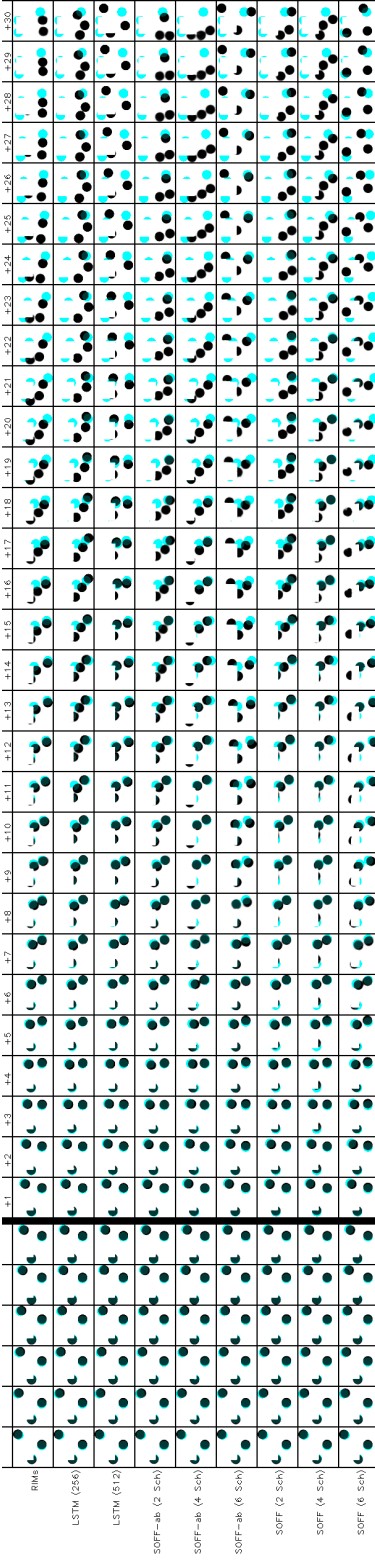

Figure 9: **Rollout for Curtain.** In all cases, the first 10 frames of ground truth are fed in (last 6 shown) and then the system is rolled out for the next 30 time steps. In the predictions, the transparent blue shows the ground truth, overlaid to help guide the eye.

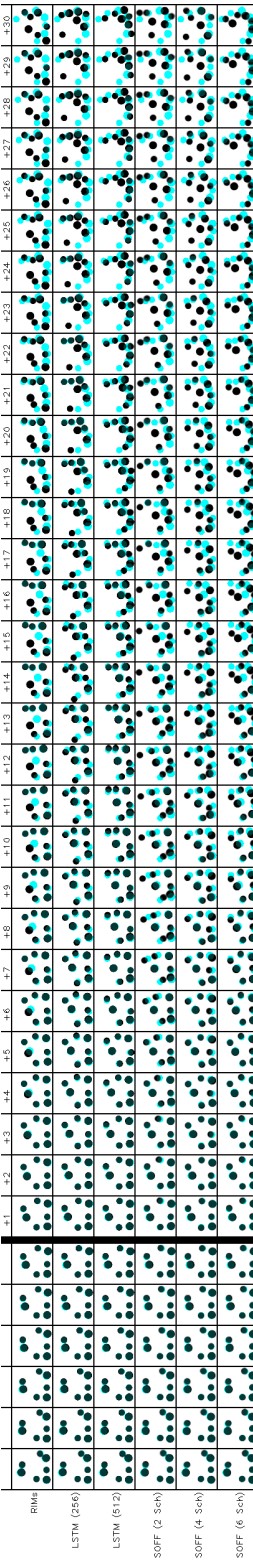

Figure 10: **Rollout for 678Balls.** In all cases, the first 10 frames of ground truth are fed in (last 6 shown) and then the system is rolled out for the next 30 time steps. In the predictions, the transparent blue shows the ground truth, overlaid to help guide the eye.

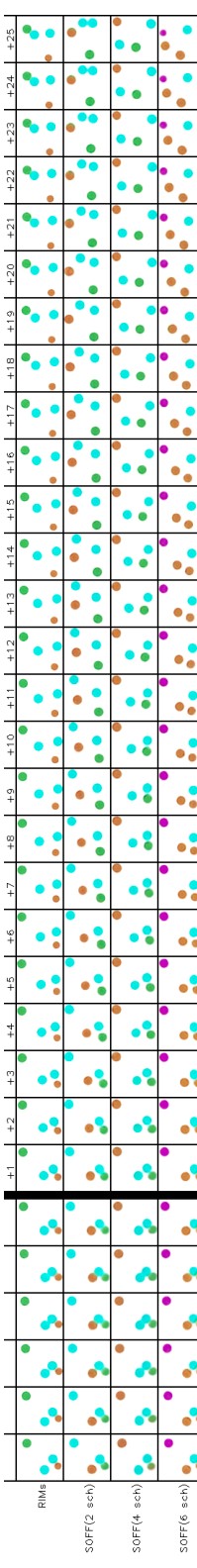

Figure 11: **Rollout for Colored 4Balls.** In all cases, the first 10 frames of ground truth are fed in (last 6 shown) and then the system is rolled out for the next 25 time steps.

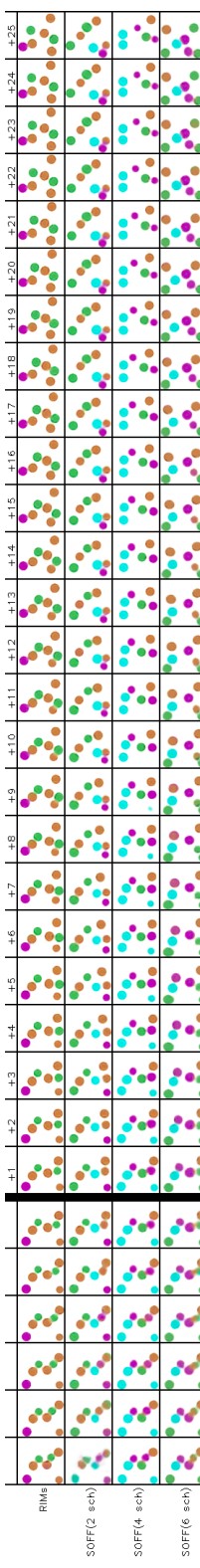

Figure 12: **Rollout for Colored 678Balls.** In all cases, the first 10 frames of ground truth are fed in (last 6 shown) and then the system is rolled out for the next 25 time steps.

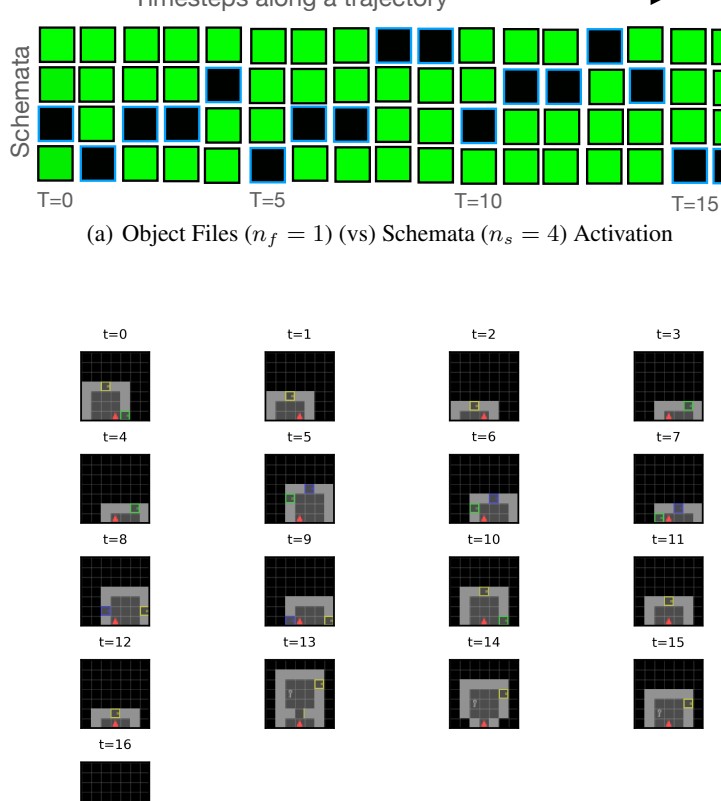

(a) Object Files ($n_f = 1$) (vs) Schemata ($n_s = 4$) Activation

(b) Agent view in the environment

Figure 13: **BabyAI-GotoObjMaze Trajectory** In this environment, the agent is expected to navigate a 3x3 maze of 6x6 rooms, randomly inter-connected by doors to find an object like "key". Here we use only one object file, but different number of schemata (4 in this example). If we look at the object files (vs) schemata affinity, schema 1 is activated while close to or opening doors while schema 4 is triggered when the "key" is in the agent's view.

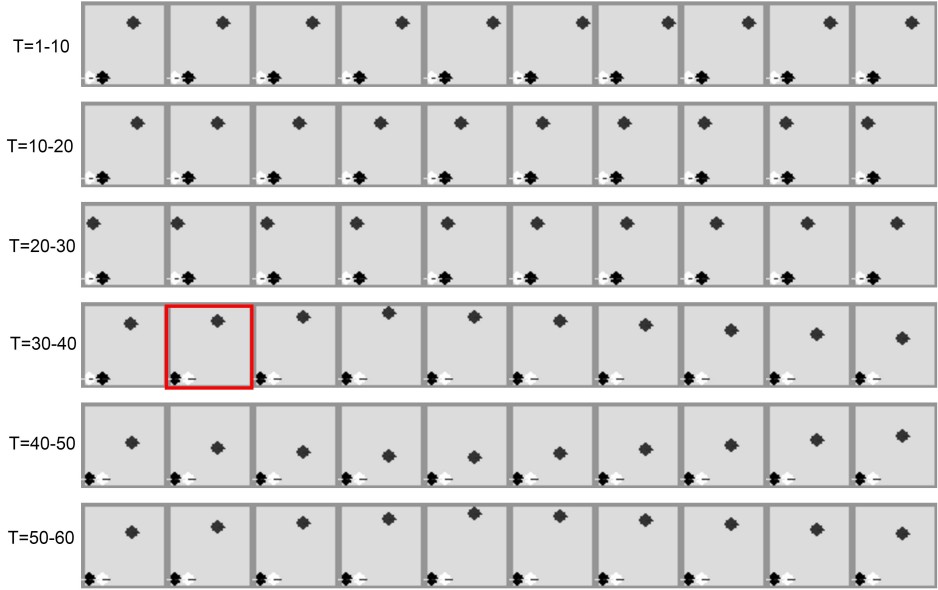

Figure 14: **Switching dynamics task.** An example ground truth trajectory, where the ball oscillates horizontally and switches to vertical oscillations after few steps indicated by the red box. The bottom left has two bulb indicators corresponding to the two dynamics.

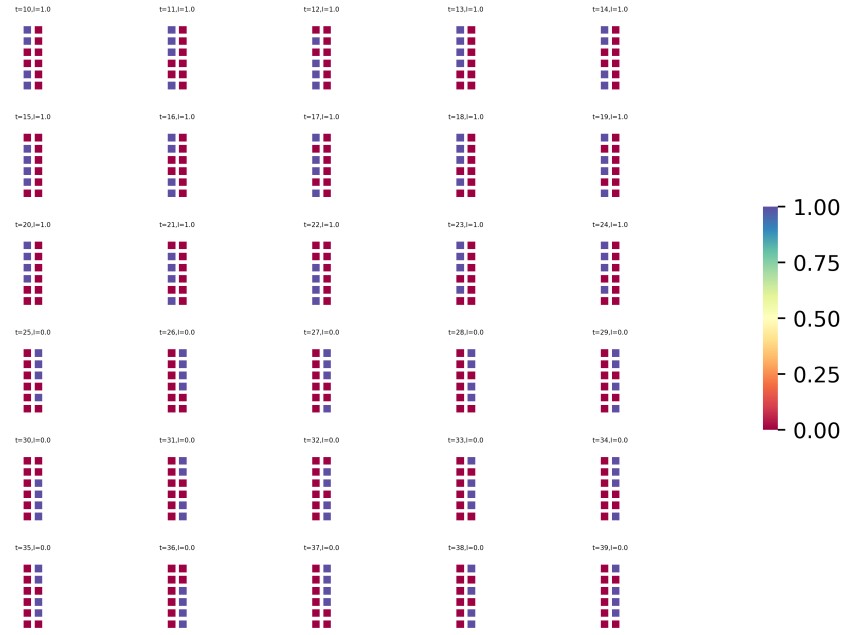

(a) Object Files ($n_f = 6$) (vs) Schemata ($n_s = 2$) for trajectory one.

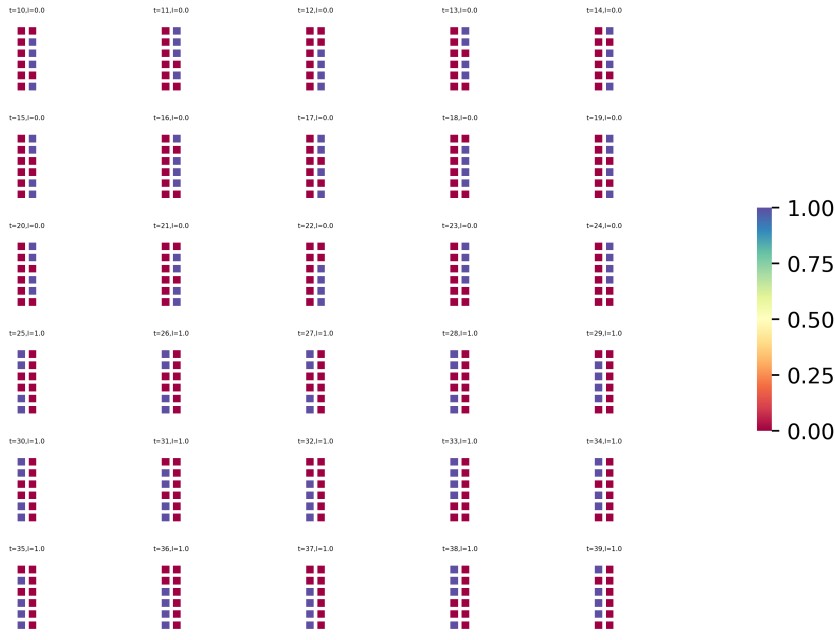

(b) Object Files ($n_f = 6$) (vs) Schemata ($n_s = 2$) for trajectory two.

Figure 15: **Switching dynamics task.** $l = 0$ denotes horizontal oscillations and $l = 1$ denotes vertical oscillations. We can clearly observe that the dynamics are being factorised into separate schemata. Schemata one is being used for vertical oscillations and schemata two for horizontal oscillations.

## G   PSEUDOCODE FOR SCOFF ALGORITHM

```python
#Assume we have defined: pos_in (number of positions in the incoming input),
    num_OF_out (number of object files on the current layer), of_dim (the
    number of hidden units in each object file), bsz (the batch size).

#Step 1: Process image with fully convolutional net.
#Step 2: Soft competition among OFs to select regions of the input
#Step 3: OFs pick the most relevant schema and update
#Step 4: Soft competition among OFs to transmit relevant information
def scoff_core(inp, h):
    inp = self.CNN(inp)
    att_inp = objectfile_input_selection(inp, h)
    h_next,schema_indices = schemata_selective_update(att_inp, h)
    h_next = h_next + objectfile_communication(h_next, schema_indices)
    return h_next

def objectfile_input_selection(inp, h):
    inp = inp.reshape((bsz, pos_in, OF_dim))
    att_inp,_ = self.inp_attn(h.reshape((bsz, num_OF_out, OF_dim)), inp, inp)
    att_inp = att_inp.reshape((bsz, of_dim*num_OF_out))
    return att_inp

def schemata_selective_update(inp_to_objfile, h):
    h_all = torch.zeros((bsz*num_OF, num_schemata, OF_dim))
    #We batch the rest of these operations over the object files.
    inp_to_objfile = inp_to_objfile.reshape((bsz*num_OF, OF_dim))
    h = h.reshape((bsz*num_OF, OF_dim))
    for j in range(num_schemata):
        h_all[:,j] = self.gru_lst[j](inp_to_objfile, h)
        key[:, j] = self.key(h_all[:,j])
    q = self.query(h)
    h_sel, ind = gumbel_softmax(torch.bmm(key, q.unsqueeze(1)))
    return h_sel.reshape((bsz,num_OF,-1)), ind.reshape((bzs,num_OF,-1))

def objectfile_communication(h, schema_indices):
    h = h.reshape((bsz, num_OF_out, of_dim))
    q = self.comm_query(h)[:,schema_indices]
    k = self.comm_key(h)[:,schema_indices]
    v = self.comm_value(h)[:,schema_indices]
    h_att,_ = self.comm_attn(q, k, v)
    h_att = h_att.reshape((bsz, num_OF_out*of_dim))
    return h_att
```

Figure 16: The core SCOFF module for a single recurrent step using GRU independent dynamics.
We show how the attention scores from attention are used to re-weight the input to the different object
files (Step 2 in the box). Then different object files select particular schemata (Step 3 in algorithm),
and then different object files transmit relevant information (Step 4 in algorithm).

