# OpenReview forum: "Factorizing Declarative and Procedural Knowledge in Structured, Dynamical Environments"
_ICLR.cc/2021/Conference — ICLR 2021 Poster_

### Official Review · AnonReviewer2 · 2020-10-28
**Nice but no attention over input, and other issues**

**Rating:** 7
**Confidence:** 4

**Review:**

Brief summary of your review:

I like the main idea of having an “active memory” where each slot can choose which operation to perform. It is similar to the notion of variables and functions, which should be advantageous for systematic generalization, which is one of the most significant issues of current neural networks.

The paper focuses on the visual domain and is motivated by extracting objects and their behavior. However, the model does not use attention over the input. The only way that the proposed model can use different object files for different instances of the same object type is to have separate schemata (because all other weights are shared), which defeats the purpose.

Thus, sadly, I must reject the paper.

Review:

The authors focus on a very important question of current neural networks: systematic generalization. Their method is interesting: they factorize modeling the scene of objects into object files (memory slots) and schemata (different learned operations). This is related to but different from memory networks, which use a single controller (instead of the multiple schemata), and also to routing networks, which use a single state (instead of multiple object files) and multiple sets of weights describing the different operations.

The authors introduce their method as a way of modeling multiple objects, some of which may share behavior. This can be seen clearly from Figure 1, but also the tone of the whole paper is organized around this goal. However, the way the input image is handled makes this very implausible. Specifically, the problem is in step 2 of Algorithm 1:

The object files compete for the input. In terms of the Pacman example, this implies that multiple moving ghosts, each with its own OF, would have to compete with each other to see the input, which makes no sense: all of them should see the change in their respective object.

The other, perhaps even more significant issue regarding input handling is that there is no attention over the input image. I cannot see how this architecture can focus on a different subset of the input corresponding to different objects. To focus on objects separately, they necessarily have to have different schemata, because the rest of the weights are shared between OF. That is the only way of having a different transformation of the input (assuming the same initial hidden state). This defeats the purpose of schemata, which is to reuse computation when possible. The current setup would make more sense in other input domains, for example for textual input, where different “objects” are not represented by different subsets of a single input z_t.

In Figure 3 the authors analyze which schema the model uses for a single object slot. The result is very nice and it shows that the model learns to use different schema for different types of motion/rooms. However, especially because of the aforementioned issues, it is unclear how the model uses the object files. I would like to see a plot similar to Figure 3, just showing which OF is used. The experiment could be to add/remove objects, and see the number of object files, or to corrupt specific files and see which object becomes unpredictable, thus identifying which OF corresponds to which object.

I still consider the paper interesting. But the issue above must be fixed, e.g., by changing the tone of the paper and shifting away from focusing on objects or switching the attention in stage 2 to attend to image regions and not over the object files.

More minor issues/questions:

In step 3, the attention is based on comparing the updated state to the old one. Why does this make sense? Is there an intuition behind this? Wouldn’t it make more sense to have fixed keys identifying the schemata?

In step 4, why do the queries use state from the previous time step as opposed to the current?

On page 4, step 2, $\kappa_k$ is mentioned, however, it should be $\kappa_t$, as $\kappa$ is independent of the object file.

On page 4, saying that Gumbel softmax “softens the output” is misleading. The goal is to have a hard, but differentiable output instead of a soft one.

In the related work section, you compare to CNNs. The same argument of weight sharing can also work for RNNs. However, differently from SCOFF, CNNs can actually “attend” to different parts of the input image, thus in theory focus on different objects differently.

Related work: Routing networks are [1,2,3] are also related. Page 2 and 4: GRU is a variant of vanilla LSTM with forget gates (Gers et al, 2000).

On page 5, the last paragraph, it is described how the output is produced from the OFs. However, it is not clear how this attention works. What is the query? Is it a transformed state? If the transformed stats are concatenated to form the query, then it is not symmetric anymore, so it will not “ensure exchangeability of OFs”. It can just focus on a single OF, based on its index.

In the experiments section, page 6, for the “Single object with switching dynamics” why does the network need markers? Can’t it infer the schema to use just from the motion itself?

Page 6, last paragraph, it should refer to Figure 3 instead of Figure 4.

On page 7, “Multiple objects with multiple dynamics”, the authors use the dataset proposed by “Van Steenkiste et al.”, and claim that they outperform their baselines. But they do not compare to the original method proposed in the same work. How does it compare?

Could the 1.3 lines long appendix E be a footnote instead?

[1] Routing Networks: Adaptive Selection of Non-linear Functions for Multi-Task Learning

[2] Kirsch et al: Modular Networks: Learning to Decompose Neural Computation

[3] Chang et al: Automatically Composing Representation Transformations as a Means for Generalization

******************

After rebuttal:

Glad to see this was just an error in the decription of the algorithm! Score increased from 4 to 7. We'd even increase our score to 8 if the authors added an analysis similar to the one of Figure 3, just showing which OF is used. As stated in the original review: The experiment could be to add/remove objects, and observe the number of object files, or to corrupt specific files and see which object becomes unpredictable, thus identifying which OF corresponds to which object.

---

> ### Author Response · Authors · 2020-11-11
> **Response: Our model does have spatial attention!**
>
> Thank you again for your thoughtful review. We thank the reviewer for the positive and constructive feedback. We appreciate that the reviewer finds our method interesting.
>
> >  “The object files compete for the input. In terms of the Pacman example, this implies that multiple moving ghosts, each with its own OF, would have to compete with each other to see the input, which makes no sense: all of them should see the change in their respective object.”
>
> >  “there is no attention over the input image ... how this architecture can focus on a different subset of the input corresponding to different objects.”
>
> > “Model does not use attention over input ... CNNs can actually attend to different parts of the input image”
>
>
> **We screwed up the description of our model.  We have updated our paper with the necessary corrections to make the explanation of this part of the model.** There are spatial attentional competitions, separate for each region of the input.    In brief, the CNN produces a 64x64 image representation and a separate key for each region of the image (which includes information about the content as well as a positional encoding). The competition in step 2 of our algorithm takes place independently for each of these regions. We apologize profusely;  what we wrote up was a simplification for the case of a single object file (where no competition is necessary) or symbolic input (see addition task in the appendix).
>
> >  “ The only way that the proposed model can use different object files for different instances of the same object type is to have separate schemata (because all other weights are shared), which defeats the purpose.”
>
> There is no competition among the object files to access a schema. Thus, multiple object files can use the same schema. Perhaps the reviewer's "separate schemata" comment was a consequence of our failing to explain spatial attention.
>
> > “Routing networks [1,2,3] ... related. Page 2 and 4: GRU is a variant of vanilla LSTM with forget gates (Gers et al, 2000).”
>
> We thank the reviewer for pointing it out. We have updated the paper citing the relevant work.
>
> >  “Plot about which object file is being used”
>
> We note that in that plot, only one single object file is used (since there’s only one entity).
>
> >  $\kappa_{k}$ is mentioned, however, it should be $\kappa_{t}$ i.e., independent of the object file.
>
> We thank the reviewer for pointing it out. We have fixed it in the most recent version.
>
> >“Gumbel softmax softens the output is misleading”
>
> We agree with the reviewer and we have made the required change.
>
> > On page 5, the last paragraph, it is described how the output is produced from the OFs. However, it is not clear how this attention works. What is the query? Is it a transformed state? If the transformed stats are concatenated to form the query, then it is not symmetric anymore, so it will not “ensure exchangeability of OFs”. It can just focus on a single OF, based on its index.
>
> The transformed state is concatenated to form a set of slots over which attention operates to decode the global state. The use of attention ensures exchangeability.  Had we used a fully connected net as a decoder, exchangeability would be an issue. The query is a learned  parameter vector which is used to select from the object files for the output from the SCOFF model.
>
> > "In the experiments section, page 6, for the 'Single object with switching dynamics' why does the network need markers? Can’t it infer the schema to use just from the motion itself?"
>
> The training performance for the LSTM baseline was worse without the visual markers, as the interval of switching between two dynamics is stochastic and hence it makes the prediction problem difficult, but conditioning on the visual markers makes the problem deterministic.
>
>
> > “I like the main idea of having an 'active memory' where each slot can choose which operation to perform. It is similar to the notion of variables and functions, which should be advantageous for systematic generalization, which is one of the most significant issues of current neural networks.“
>
> We very much agree with the reviewer, and we thank the reviewer for pointing it out.  One can think of different schemata as analogous to classes in programming language, and think of different object files as analogous to different objects (i.e. we can have multiple objects which use the same class). It also opens the avenues for future research.

---

> > ### Author Response · Authors · 2020-11-12
> > **Updated Manuscript**
> >
> > Dear. Reviewer,
> >
> > Thanks again for your time, and for your valuable feedback.
> >
> > We have uploaded a new version with a description of spatial attention.
> >
> > Changes:
> >
> > - Changes in Step 1 and Step 2 of the algorithm box.
> > - Changes in Step 1 and Step 2 of the section 2.1.
> > - Cited the papers in the related work section.
> >
> > We thank the reviewer again, for their feedback. It has helped us to improve the writing of the paper. We really appreciate it.

---

### Official Review · AnonReviewer4 · 2020-10-28
**Very interesting work with good experiments.**

**Rating:** 8
**Confidence:** 4

**Review:**

The authors propose SCOFF, a novel architectural motif, one with memory, which, as they describe, can serve as a drop-in for an LSTM or GRU within any architecture. It is inspired by the notion that when modeling a structured, dynamic environment (such as one with objects moving around), one must keep track of both declarative knowledge and procedural knowledge. They propose that these two types of knowledge be factored, creating an architecture consisting of "object files" (OF) whose evolution is governed by input, all objects, and  "schemata" which can be selectively applied to each OF.

They evaluate SCOFF along several axes:
(1) Does SCOFF successfully factorize knowledge into OFs and schemata?
(2) Do the learned schemata have semantically meaningful interpretations?
(3) Is the factorization of knowledge into object files and schemata helpful in downstream tasks?
(4) Does SCOFF outperform state-of-the-art approaches?
To do this, they use several video prediction tasks as well as an RL task.

This is a very interesting paper, with a natural, novel motif. It is well-written -- the motif has several important attributes, and one can quickly come to understand them (though perhaps a more involved diagram, like Figure 2 but showing what parameters come into play where, might be useful). The experiments appear to be carefully done, with considerable effort, and they put forth interesting evidence towards an affirmative in each of the above four questions.

While the evidence put forth is useful, I do think there is considerable follow-up work needed to really demonstrate the efficacy of this system. In the realm of video prediction, the tasks considered are fairly simple, and certainly what is of true interest is video prediction much closer to the real world. With this, there are a wide variety of techniques and benchmarks (https://arxiv.org/abs/1804.01523 and its follow-ons come to mind as useful to quickly try). Much recent work has deferred the task of obtaining a reasonable encoding from/decoding to real-world images and assumed it in order to make progress on predicting the future within a fixed encoding (https://arxiv.org/abs/1612.00222, https://arxiv.org/abs/2002.09405, https://arxiv.org/abs/1806.08047) and have with them benchmarks to which the authors' method could be adapted. Certainly for some of these, SCOFF could be a subcomponent that augments these methods. I think these more complex future predictive tasks could better stress-test the structure of how information passes through SCOFF. Related, many of the experiments rely on the encoding/decoding provided by [Van Steenkiste 2018], which showed considerable success on the environments used, and it would be interesting to see how crucial that is.

I recommend acceptance. The paper is interesting, well-written, and the experiments are useful. While I look forward to more definitive demonstrations of the utility of this approach, I do think that the amount done is considerable and warrants publication, and these important follow-ups would take a great deal more effort and are for future work.

A more minor comment, I think more details could be given for the RL task, both in model implementation and in exactly how the test task is specified -- apologies if I missed but I only see train environment details in the appendix.

Minor, wrong "two"/"to" in "Single object with switching dynamics" experiments description.

---

> ### Author Response · Authors · 2020-11-11
> **Thanks for your feedback**
>
> We thank the reviewer for their feedback and their generally positive assessment of our work. We will add the details for RL experiments in the appendix, and we agree using such ideas for real world video prediction tasks is an interesting avenue for future research.

---

### Official Review · AnonReviewer5 · 2020-11-05
**Strong inductive bias for specific tasks**

**Rating:** 6
**Confidence:** 4

**Review:**

This paper proposes a new type of recurrent neural network architecture called schema / object-file factorization (SCOFF). This model contains multiple weight-sharing GRU cells. The input information is fed into each GRU cells through an attention layer. The output information is fetched from these GRU cells and mixed with another attention layer. The model is tested on several intuitive physics benchmarks and basic reinforcement learning environment. This model demonstrates superior performance than other modular RNN architectures such as RIM on specific tasks.

+ves:

+ Overall, the paper is well written. Section 2 clearly explains the proposed model. Section 3 systematically compared the proposed model against other RNN architectures.

+ All experiments results covered in detail including hyperparamters and experimental setting.


Concerns:

- This paper uses large amount of neuroscience terminology and vague concepts, which are just renaming of existing concepts. This is not novelty or contribution. And it is unnecessary and inappropriate for the conference publication.


- The model is simply doing attention + weight sharing RNNs + attention. The only difference from Recurrent Independent Mechanisms (RIMs) is that the modules has shared weights - an inductive bias is be useful for specific tasks. The novelty is weak.


- The proposed model on intuitive physics experiments shows slightly better performance than RIM. However, all these experiments have extreme setting, which is for the model's inductive bias that all objects follow exact same rules and contain full input information. It's obvious that the proposed model will perform worse than RIM when not all objects sharing same rules.


- For these module based RNN models, it is necessary to show that the attention layer is functioning as expected. Therefore, I would suggest that the paper add visualization of the attention layers.

=====POST-REBUTTAL COMMENTS========

I thank the authors for the response. All my concerns are addressed. I will increase my score to 6.

---

> ### Author Response · Authors · 2020-11-10
> **SCOFF is sharing parameters, but in a dynamic, state-dependent manner**
>
> We thank the reviewer for the positive and constructive feedback.
>
> > 1. “This paper uses a large amount of neuroscience terminology and vague concepts, which are just renaming existing concepts. This is not novelty or contribution. And it is unnecessary and inappropriate for the conference publication.”
>
> The terminology (object files, schemata, procedural and declarative knowledge, types and tokens) are foundational concepts of cognitive science from the 1980s. Back propagation and neural networks arose from the same field. (And in fact, Dave Rumelhart, who invented back propagation, also helped popularize the term 'schemata'.) We don't claim novelty, but we feel it is important to ground our work in the historical terminology.  We are not aware of corresponding terminology in use in machine learning, but we would appreciate pointers if it exists and if it has precedence over the cognitive science terminology. For some historical grounding, see
> https://en.wikipedia.org/wiki/Type%E2%80%93token_distinction
> https://en.wikipedia.org/wiki/Schema_(psychology)
> https://en.wikipedia.org/wiki/Procedural_knowledge
> https://en.wikipedia.org/wiki/Feature_integration_theory
>
> > 2. “The model is simply doing attention + weight sharing RNNs + attention. The only difference from Recurrent Independent Mechanisms (RIMs) is that the modules has shared weights - an inductive bias is be useful for specific tasks. The novelty is weak.”
>
> The difference from RIMs is that in RIMs, each module has its own separate parameters.  In SCOFF, modules (object files) dynamically select parameters from a common pool in a state-dependent manner.  SCOFF shares the notion of modularity with RIMs but is far more dynamic.  This dynamic assignment of parameters to modules results in exchangeability of modules, and has an analogy with the classes vs. objects distinction in programming languages.
>
> Our experiments show that this property is essential, as n_s = 1 (using just a single schema) has poor results, and we show strong results with a variety of n_s when n_s > 2.  It is actually a good point that we should have emphasized this much better in the paper, and we will update the paper to reflect this.
>
> We ran on Atari with a variety of n_s values to illustrate this point.  We report scores on 5 different games (higher mean is better):
>
> Games                     = Alien, Amidar, Assault, Asterix, MsPacMan
>
> - LSTM                               = 16920, 3944, 40874, 572150, 7184
> - LSTM (4x parameters) = 17232, 3985, 42134, 563238, 6832
> - N_f = 1, n_s=2                 = 18121, 4200, 37000, 621210, 7232
> - N_f = 1, n_s=4                 = 19231, 4182, 41232, 621210, 7900
> - N_f = 4, n_s = 1               = 8123,   2312, 31294, 487340, 5621
> - N_f = 4, n_s = 4               = 17241, 4231, 39583, 643427, 7893
>
> Here, n_f refers to number of object files, and n_s refers to number of schemata. Here, We  ran on Atari with Recurrent IQN, where SCOFF is a drop-in replacement for the LSTM. We achieve good results with n_s=2 and n_s=4, yet consistently very poor results with n_s=1 (which is where all object files share the same parameters).
>
> > 3. “However, all these experiments have extreme settings, which is for the model's inductive bias that all objects follow exact same rules and contain full input information. It's obvious that the proposed model will perform worse than RIM when not all objects sharing same rules.”
>
> Can you clarify what you mean here?  If all objects sharing the same rules worked well, as you claim, then using 1 schema (n_s=1) would perform well.  However it performs very poorly (see Atari results above).  Also, there is an attentional competition for the input, so modules do not obtain "full input information".
>
> > 4. “For these module based RNN models, it is necessary to show that the attention layer is functioning as expected.”
>
> We totally agree with the reviewer. In order to study this, in the paper we considered video scenes in which a single object, starting in a random location, has dynamics that cause it to either (a) accelerate in a particular direction, (b) move at a constant velocity in a particular direction, or (c) take a random walk with a constant velocity. Following training, SCOFF can predict trajectories after being shown the first few frames. It does so by activating a schema that has a one-to-one correspondence with the three types of dynamics  (Figure 3, left panel), leading to interpretable semantics and a clean factorization of knowledge. We also study similar problem with multiple dynamics, and the  visualization is exactly what’s shown in Figures 3,4,6.
>
> Please let us know if there is anything we can address for you to increase your score.

---

> > ### Author Response · Authors · 2020-11-16
> > **Anything else you'd like us to respond to?**
> >
> > Hello,
> >
> > We thank the reviewer for their feedback and valuable comments.
> >
> > Since the first phase of response period is closing soon, if you have time and could indicate if there are any other concerns of yours which we have not addressed, we'd be happy to take a look.
> >
> > Thanks for your time.

---

> ### Author Response · Authors · 2020-11-17
> **Thanks for your prompt reply and increasing score.**
>
> Understanding the visual world requires interpreting images in terms of distinct independent physical entities. These entities have persistent intrinsic properties, such as a color or velocity, and they have dynamics that transform the properties. We explored a mechanism that is able to factorize declarative knowledge (the properties) and procedural knowledge (the dynamics). Using attention, our SCOFF model learns this factorization into representations of entities—OFs—and representations of how they transform over time—schemata. By applying the same schemata to multiple OFs, SCOFF achieves systematicity of prediction, resulting in significantly improved generalization performance over state-of-the-art methods. It also addresses a fundamental issue in AI and cognitive science: **the distinction between types and tokens**.
>
> For example, if we had a system with 3 balls distinct in color but sharing the same dynamics, RIMs would need to learn different dynamics for each ball, whereas SCOFF could learn to reuse the parameters between the different modules, even though the balls themselves have different states.
>
> > "empirical results are not convincing."
>
> On the point of experimental results, we've improved this some in the rebuttal by adding Atari results.  Nonetheless, are any other results / experiments / settings that you'd be particularly interested in seeing (particularly something where LSTM or RIMs already is used)?  If so we might be able to run it since SCOFF follows the same input/output interface.

---

### Official Review · AnonReviewer1 · 2020-11-06
**Genaralizing RIMs and GRUs by splitting schema memory vs. per-frame feature learning. Good idea. Needs work.**

**Rating:** 5
**Confidence:** 3

**Review:**

The motivation and the proposal for splitting the schema from the procedural (representational) block makes sense. This is a good idea. A the authors build on top of RIMs, which have shown reasonable ways to model dynamical systems. However the paper itself needs to be improved and we need to evaluate the model more before publication.

Firstly, the proposal that SCOFF is a direct alternative for LSTM or GRU and showing that it beats them is not entirely correct. SCOFF comprises of a GRU with a sequence of CNNs operations i.e., its doing more than what a GRU does? What exactly is that? And so when proposing evaluations it is expected that compared to GRU, SCOFF does better (Fig 5). A more valid comparison would be GRU+some standard CNN-style feature learning vs. SCOFF. Its not entirely clear how to do this -- and needs thinking. Secondly, while on Fig 5, the errors cannot be reported as a ratio with respect to GRU because this would miss the true error values; and we cannot know if there is significant difference here (especially with RIMs.vs SCOFF).
Next, how is question 3 evaluated here? What downstream task is being considered? Am I missing something?
Next, how doe we interpret the error change here i.e., what does 0.1 change in error mean here for better understanding where things are changing?

Presentation of the paper needs a lot of improvement:
Algorithm 1 in the Table needs better clarity. Firstly, a bulk of the notation in the SCOFF presentation is very confusing. Its hard to parse what is going on in each stage. The description in the steps is helpful but the motivation and sequence of operations within each step needs better explanation.

---

> ### Author Response · Authors · 2020-11-10
> **SCOFF vs. GRU Baselines are Fair Comparisons**
>
> We thank the reviewer for the positive and constructive feedback.
>
> > 1. “SCOFF comprises a GRU with a sequence of CNNs operations i.e., its doing more than what a GRU does? What exactly is that? … A more valid comparison would be GRU+some standard CNN-style feature learning vs. SCOFF“
>
> When we compare SCOFF to a GRU baseline, we use the same convolutional encoder to encode the input image for both.  If one views a recurrent network as an interface: h[t] ← recurrent(h[t-1], x[t]), then we use SCOFF as a drop-in replacement which follows the exact same interface as the GRU baseline.  There is never a case where SCOFF gets extra convolutional layers to process the input image or somesuch, which we agree would be deeply unfair.
>
> Regarding how SCOFF improves on GRUs, we want to note that SCOFF is a modular architecture which factorizes knowledge about the entities and dynamics of different entities. Moreover, it generally reduces the number of parameters and has the same number of units as a GRU baseline.
>
> > 2. “Next, how is question 3 evaluated here? What downstream task is being considered?”
>
> In order to evaluate the performance of a proposed model on a downstream task,  we study the performance of the proposed model in the IntPhysics Benchmark. Modeling a physical system, such as objects in a world obeying laws of gravity and momentum, requires factorization of state (object  position, velocity) and dynamics. All objects must obey the same laws while each object must maintain its distinct state.  We used the Intuitive Physics Benchmark \citep{riochet2019intphys} in which balls roll behind a brick wall such that they are briefly occluded. We test  three forms of unrealistic physics: balls disappearing behind the wall (O1 task), balls having their shape change for no reason (O2 task), and balls teleporting (O3 task). The Benchmark has three subsets of experiments, and we chose the challenging subset with significant occlusions. As Table 1 indicates, SCOFF significantly outperforms two competitors on all three tasks. We also evaluate the performance of the proposed method in RL paradigm (with single object file, and multiple schemata). See page 5 under the paragraph “Single object with switching dynamics in an RL paradigm.”
>
>
> > 3. “the errors cannot be reported as a ratio with respect to GRU“
>
> We can assure you that the error value differences are significant in absolute terms, and that the ratios are used to simplify the presentation of results on different tasks (where the errors are scaled differently).
> Nonetheless we  report these values here as well:
> - For 4 balls: BCE error LSTM: 0.73, RIMs: 0.52, SCOFF: 0.37 (BCE) After 30 time-steps.
> - For Curtain : BCE error LSTM: 0.72, RIMs: 0.41, SCOFF: 0.29 (BCE) After 30 time-steps.
>
> > 4. “Presentation of the paper needs a lot of improvement: Algorithm 1 in the Table needs better clarity”
>
> Do you have any more specific feedback on presentation or the algorithm block?  We want to make the presentation as clear as possible, and we note that Reviewer 4 and Reviewer 5 found the paper to be well written.  Thus it would be helpful to know what specific aspects you found unclear or could benefit from further exposition.  Thanks very much for your time.

---

> > ### Author Response · Authors · 2020-11-13
> > **Updated manuscript**
> >
> > Dear reviewer,
> >
> > We have updated our manuscript. If there’s any aspect you think we could explain better, please do not hesitate to leave us comment. Let us know if there's anything else we can do improve it further and hence increase your score.
> >
> > Thank you.

---

> > > ### Author Response · Authors · 2020-11-17
> > > **Anything else you'd like us to respond to?**
> > >
> > > Hello,
> > >
> > > We thank the reviewer for their feedback and valuable comments.
> > >
> > > Since the first phase of response period is closing soon, if you have time and could indicate if there are any other concerns of yours which we have not addressed, we'd be happy to take a look.
> > >
> > > Thanks for your time.

---

### Comment · AnonReviewer2 · 2020-11-17
**Glad to see this was just an error in the decription of the algorithm! Score increased from 4 to 7 (8 under condition).**

Glad to see the main problem was just an error in the decription of the algorithm!  It makes much more sense now. We are increasing our score from 4 to 7.

We'd even increase our score to 8 if the authors added an analysis similar to the one of Figure 3,  just showing which OF is used. As stated in the original review: The experiment could be to add/remove objects, and observe the number of object files, or to corrupt specific files and see which object becomes unpredictable, thus identifying which OF corresponds to which object.

---

### Decision · Program_Chairs · 2021-01-07
**Final Decision**

**Decision:**

Accept (Poster)

**Comment:**

This paper proposes a modular RNN architecture called SCOFF. The work was inspired by cognitive science(object file and schema) and was built upon previous work RIMs. The method is validated on tasks having multiple objects of the same type.

Pros:
- It addresses an important problem in DNN -- systematic generalization.
- The proposal makes sense and is more flexible than RIM.
- Experimental results outperform baselines.

Cons before rebuttal:
- The presentation of the algorithm is not very clear due to some confusing notations and missing details of algorithm steps.
- The comparison with baselines might not be fair due to extra parameters.
- The novelty is limited, because the only difference from RIM is weight sharing.

The reviewers raised concerns listed in Cons. The authors successfully addressed concerns: they indicated that the comparison was fair with the same input to both; SCOFF is more flexible than RIM, and there is spatial attention to input.
The authors added the missing details in the revised version.

All reviewers agree that the problem is important and the idea is interesting.  Since the authors' rebuttal was very helpful in clarifying the questions raised, I recommend accept.